# Impact Localization System of CFRP Structure Based on EFPI Sensors

**DOI:** 10.3390/s25041091

**Published:** 2025-02-12

**Authors:** Junsong Yu, Zipeng Peng, Linghui Gan, Jun Liu, Yufang Bai, Shengpeng Wan

**Affiliations:** 1Key Laboratory of Nondestructive Testing, Ministry of Education, Nanchang Hangkong University, Nanchang 330063, China; 2308085408014@stu.nchu.edu.cn (Z.P.); 2308080300007@stu.nchu.edu.cn (L.G.); j132004@163.com (J.L.);; 2Key Laboratory of Opto-Electronic Information Science and Technology of Jiangxi Province, Nanchang Hangkong University, Nanchang 330063, China; 3School of Electrical Engineering, Shanghai Dianji University, Shanghai 201306, China

**Keywords:** structural health monitoring, impact localization, EFPI, energy–entropy ratio, CNN-BIGRU-Attention

## Abstract

Carbon fiber composites (CFRPs) are prone to impact loads during their production, transportation, and service life. These impacts can induce microscopic damage that is always undetectable to the naked eye, thereby posing a significant safety risk to the structural integrity of CFRP structures. In this study, we developed an impact localization system for CFRP structures using extrinsic Fabry–Perot interferometric (EFPI) sensors. The impact signals detected by EFPI sensors are demodulated at high speeds using an intensity modulation method. An impact localization method for the CFRP structure based on the energy–entropy ratio endpoint detection and CNN-BIGRU-Attention is proposed. The time difference of arrival (TDOA) between signals from different EFPI sensors is collected to characterize the impact location. The attention mechanism is integrated into the CNN-BIGRU model to enhance the significance of the TDOA of impact signals detected by proximal EFPI sensors. The model is trained using the training set, with its parameters optimized using the sand cat swarm optimization algorithm and validation set. The localization performance of different models is then evaluated and compared using the test set. The impact localization system based on the CNN-BIGRU-Attention model using EFPI sensors was validated on a CFRP plate with an experimental area of 400 mm × 400 mm. The average error in impact localization is 8.14 mm, and the experimental results demonstrate the effectiveness and satisfactory performance of the proposed method.

## 1. Introduction

Compared with traditional materials, CFRP offers advantages such as low mass, high specific strength and modulus, low creep, and excellent corrosion resistance. These characteristics make it widely used in fields such as aerospace and rail transit [1,2]. However, during the manufacturing, transportation, and service processes, CFRP structures are prone to damage from collisions with other objects. Such damage can significantly compromise the structural integrity of CFRP, reducing its bearing capacity and reliability. Moreover, these damages are often undetectable to the naked eye [3,4]. Therefore, developing health monitoring technology with impact localization ability is of significant engineering practical importance for the maintenance of CFRP structures.

Numerous studies explored the use of PZT and FBG sensors for impact localization in composite and metallic structures [5]. Yan et al. introduced a technique that involves comparing time-frequency domain features between the impact point and reference impact point using the Pearson correlation coefficient to achieve impact localization [6]. Qi et al. developed a methodology that extracts wavelet packet features from impact signals and determines the impact location by minimizing the disparity between the impact signals and reference signal through the application of a search algorithm [7]. However, methods relying on signal difference comparisons exhibit limited resilience to signal interference. Consequently, machine learning has been widely adopted in impact localization to improve generalization capabilities. Shenoy et al. extracted impact signal features from the frequency spectrum and power spectral density using principal component analysis and applied these features for impact localization on composite sandwich plates through densely based neural networks [8]. Li et al. analyzed the time and frequency domain characteristics of impact signals, subsequently employing these features as input variables for support vector regression and artificial neural networks to predict the impact locations [9]. The effectiveness of machine learning-based methods depends on the sensitivity of the extracted eigenvalues to the impact location. Additionally, impact localization techniques that utilize the arrival time of impact signals are frequently employed for localizing the impact point. Zhang et al. utilized the time difference matrix matching method to achieve impact localization on composite material structures and improved localization accuracy through edge precision optimization [10]. Jang et al. adopted the triangulation method using time-of-arrival data derived from amplitude thresholds to determine the impact location [11]. However, these methods necessitate a high-speed FBG interrogator to capture the arrival time differences between signals. The precision of determining these time differences depends on the chosen method for extracting the signal arrival times.

Fiber optic sensors, in contrast to conventional PZT sensors, present several advantages such as small size, light weight, high sensitivity, and resistance to electromagnetic interference. Meanwhile, FBG sensors and their high-speed demodulation systems entail high costs and have different sensitivities to Lamb waves caused by impact loads in different directions [12]. Due to the advantages of fiber optic sensors, as well as the symmetry of sensor structures and the simplicity and low cost of sensor manufacturing and demodulation systems, EFPI sensors are widely used in pipeline corrosion detection [13], ocean monitoring [14], biomedical purposes [15], and sound source localization [16,17,18,19].

Therefore, this paper presents an impact localization method for CFRP structures based on EFPI sensors and CNN-BIGRU-Attention model. Impact signals are detected by EFPI sensors and collected via high-speed demodulation technology. This method involves extracting the arrival time of impact signals measured by EFPI sensors at different locations using the energy–entropy ratio endpoint detection algorithm. Then, the time difference between the impact signals arriving at different sensors are considered as signal features. The feature dataset is divided into training, validation, and test sets. The models for impact localization are trained using the training set. The validation set is used to optimize model parameters, while the performance of the trained models are evaluated by test set. By comparing the localization results of different models, it is found that the localization results of CNN-BIGRU-Attention model is satisfactory.

## 2. Impact Localization Algorithm

### 2.1. Endpoint Detection Methods

#### 2.1.1. Short Time Energy Method

Assuming $x\left(n\right)$ is the original signal and the i-th frame signal is $x_{i}\left(n\right)$, the short time energy $E_{i}$ of $x_{i}\left(n\right)$ is defined as follows:(1)$$E_{i}=\sum_{n=1}^{L}x_{i}^{2}\left(n\right)$$

In the formula, L represents the frame length.

A complete impact signal includes noise segments and impact event segments, and the short time energy of the noise segment is less than that of the impact event segment, so the endpoint of impact event can be detected by this characteristic.

#### 2.1.2. Spectral Entropy Method

The concept of entropy was initially derived from thermodynamics to quantify the degree of disorder within a system and was later introduced into information theory to measure the uncertainty of random events. With the intersection and integration of disciplines, the application of entropy in signal detection and recognition has gained increasing significance. The disorder within a signal is positively correlated with its entropy value. Typically, noise exhibits a high entropy value.

$x\left(n\right)$ is defined as the original signal, and Fourier transform was performed on it. The spectral value of the *k*-th spectral line of the i-th frame signal after Fourier transform is represented as $Y_{i}\left(k\right)$, and its normalized spectral density value $P_{i}\left(k\right)$ is represented as follows:(2)$$P_{i}\left(k\right)=\frac{Y_{i}\left(k\right)}{\sum_{l=0}^{N/2}Y_{i}\left(l\right)}$$

In the above formula, N is the FFT length. Consequently, the short time spectral entropy value $H_{i}$ of the i-th frame signal is defined as follows:(3)$$H_{i}=-\overset{N/2}{\underset{k=0}{\sum }}\left(P_{i}\left(k\right)\times \log\left(P_{i}\left(k\right)\right)\right)$$

Compared to impact signals, the normalized spectral probability density function of noise has a more uniform distribution, so the spectral entropy of noise is larger than that of impact signals. Through this characteristic, we can detect impact signals and noise.

#### 2.1.3. Energy–Entropy Ratio Method

Compared with the traditional peak method and short time energy method, the energy–entropy ratio method combines the advantages of short time energy method and short time spectral entropy method. Although peak method and short time energy method are simple to calculate but sensitive to noise and lack robustness in non-stationary signals [20]. The short time spectral entropy method performs better in noisy environments but has high computational complexity and limited adaptability to periodic noise [21]. The energy–entropy ratio method effectively highlights the significant differences between useful signal segments and noise segments by calculating the ratio of energy envelope to entropy envelope, thereby improving the accuracy and robustness of signal endpoint detection [22]. In this study, the method was applied to provide a more reliable basis for extracting the arrival time of impact signals detected by EFPI sensors.

According to the principles of short time energy and spectral entropy methods, the energy envelope and entropy envelope of a signal segment exhibit opposite trends within the same interval. By calculating the energy–entropy ratio, significant features and differences between the useful signal segment and the noise segment can be effectively highlighted. Specifically, the short time energy–entropy ratio of a signal can be defined as follows:(4)$$EEF_{i}=\sqrt{1+\left|EL_{i}/H_{i}\right|}$$

In the formula, $EL_{i}$ is short time energy and $H_{i}\;$ is short time spectral entropy. After calculating the energy–entropy ratio, the endpoint is identified by selecting an appropriate threshold. The threshold calculation formula is expressed as follows:(5)$$T_{h}=\alpha \times (M_{e}-e_{th})+e_{th}$$
where α is a constant, $M_{e}$ is the maximum value of the energy–entropy ratio, and $e_{th}$ is the average value of the energy–entropy ratio.

### 2.2. CNN-BIGRU-Attention Model

#### 2.2.1. Convolutional Neural Network

As a widely used network model in the field of deep learning, convolutional neural network (CNN) is a feedforward neural network with a depth structure [23]. As shown in Figure 1, conventional CNN typically consists of input layer, convolution layer, pooling layer, full connection layer, and output layer [24].

The input layer is responsible for receiving and inputting the original data into the CNN. Generally, in order to optimize the calculation time, the data will be preprocessed. As the most critical component of a CNN, the convolutional layer convolves the input data to extract local features. It employs a convolution kernel to perform convolution operations on the data in a sliding window manner until all data have been processed. Additionally, the convolutional layer can effectively reduce the size of the feature map and achieve dimensionality reduction by setting appropriate strides and padding. The expression for the convolution operation is expressed as follows [25]:(6)$$y^{n}=\sum_{i=1}^{b^{n-1}}x_{i}^{n-1}\times w_{i}^{n}+a_{i}^{n}$$

In the above equation, $y^{n}$ represents the local area output value of layer n, $b^{n-1}$ represents the *b*-th channel in layer n−1, $x_{i}^{n-1}$ is the output value of the *i*-th channel of the n−1 layer (i.e., the input value of the n-layer), ∗ signifies one-dimensional convolution operations, $w_{i}^{n}$ represents the weight coefficient of the *i*-th channel of the n-layer, and $a_{i}^{n}$ represents the bias of the i-th channel of the n-layer.

The convolution layer, as the most critical component of a convolutional neural network (CNN), convolves the input data to extract local features. It selects a convolution kernel to perform the convolution operation on the data through a sliding window approach until all data have been processed. By setting the appropriate stride length and padding, the size of the feature map can be effectively reduced, and the purpose of dimension reduction can be achieved.

Following the convolution layer, the pooling layer plays a crucial role in refining the feature vectors obtained from the convolution operation through down-sampling. This process reduces the feature dimension and enhances the operational speed of the CNN. Commonly used pooling operations include maximum pooling and average pooling [26]. Maximum pooling preserves important local features by selecting the maximum value in the pooled region, while average pooling preserves the overall information of features by calculating the average value of the pooled region. Therefore, maximum pooling emphasizes salient features, while average pooling maintains smoothness and a global perspective of the information. Figure 2 below shows a schematic diagram illustrating both pooling operations:

The main function of the fully connected layer is to recombine the features obtained after convolution and pooling operations, thereby reducing the dimension of features. Subsequently, these features are expanded into feature vectors.

The output layer is located behind the full connection layer, which is responsible for generating the final classification predictions and processing input values through a specified activation function to produce the required results. CNN has powerful feature extraction ability. By using its local perception and weight sharing characteristics, it can effectively reduce redundant information, learning parameters, and computational time, ultimately simplifying the complexity of the model.

#### 2.2.2. Bidirectional Gated Recurrent Unit (BIGRU)

The Gated Recurrent Unit (GRU) is an improved version of the recurrent neural network (RNN) [27], which aims to solve the gradient vanishing problem encountered by traditional RNNs when processing long sequences. By introducing a gating mechanism, GRU optimizes information flow, enabling it to retain crucial information and discard unnecessary details more effectively. Compared to standard RNN, GRU is generally easier to train, particularly with long sequence data. Furthermore, due to its reduced parameter count, GRU offers higher computational efficiency.

However, GRU has certain limitations. Its current output is solely dependent on the previous hidden state and the current input, making it challenging to capture future information within the sequence [28]. To overcome this, the Bidirectional Gated Recurrent Unit (BIGRU) incorporates a reverse GRU structure [29]. Compared to another variant of RNN that incorporates memory cells and gating mechanisms, known as long short-term memory (LSTM), while LSTM can effectively capture both long-term and short-term dependencies, BIGRU enhances model predictive capability by comprehensively capturing feature relationships in time series through simultaneous processing of forward and backward information [30,31]. The expanded model structure is illustrated in Figure 3:

As shown in Figure 3, the output layer H of BIGRU is determined by three parts: forward hidden information, reverse hidden information, and input *X*. C denotes the hidden information at time t, *S* denotes the activation value of neurons at time t, and → and ← denote forward propagation and backward propagation, respectively. The specific formula is as follows:(7)$$\left\{\begin{array}{c} {\vec{C}}^{t}=GRU\left(x_{t},{\vec{C}}^{t-1}\right)\\ {\overset{\leftarrow }{C}}^{t}=GRU\left(x_{t},{\overset{\leftarrow }{C}}^{t-1}\right)\\ h^{t}={\vec{w}}^{t}{\vec{c}}^{t}+{\overset{\leftarrow }{w}}^{t}{\overset{\leftarrow }{c}}^{t}+b^{t} \end{array}\right.$$
where $\vec{w}$ is the output weight of the forward GRU, $\overset{\leftarrow }{w}$ represents the output weight of the reverse GRU, and $b^{t}$ represents the offset corresponding to the hidden layer state.

#### 2.2.3. Attention Mechanism

The attention mechanism is a technology that assigns different weights to various parts according to their importance when processing input data. By calculating the correlation between elements within the input sequence, the model is able to prioritize more critical information, thereby enhancing its performance. Specifically, the attention mechanism optimizes the sequence generation process by calculating the attention weight and integrating the hidden state of the encoder with the hidden state of the decoder. This mechanism enables the model to dynamically focus on the most pertinent parts of the input sequence, improving both the accuracy and fluency of the generated results [32,33,34,35].

The BIGRU model enhances its localization prediction ability by simultaneously processing forward and backward data, enabling it to capture more comprehensive trends. In addition, the attention mechanism further enhances the performance of the model by selectively emphasizing the most relevant parts of the data. At each time step, the attention mechanism assigns higher weights to key features that may indicate significant changes in the impact location, ensuring that the model focuses on the most important information. The specific formula is as follows [36,37]:(8)$$e_{t}=u\times \tanh\left(w\times h_{t}+b\right)$$(9)$$\alpha_{t}=\frac{\exp\left(e_{t}\right)}{\sum_{j=1}^{t}e_{j}}$$(10)$$s_{t}=\sum_{t=1}^{i}\alpha_{t}\times h_{t}$$
where $e_{t}$ represents the value of the attention probability distribution determined by the BIGRU network layer output vector $h_{t}$ at moment t. u and w are weight coefficients, b is the bias coefficient, $\alpha_{t}$ is the attention probability distribution of the attention mechanism output to the BIGRU hidden layer, and $s_{t}$ is the output of the attention layer at time t.

#### 2.2.4. Sand Cat Swarm Optimization

Sand cat swarm optimization (SCSO) is a swarm intelligence optimization algorithm proposed to simulate the survival behavior of sand cats preying on prey [38]. The algorithm can effectively explore and develop the solution space by working together with multiple “sand cat” individuals in the search space. The algorithm can also flexibly adjust the search strategy according to the change in environment, so as to improve the efficiency of finding the optimal solution. At the same time, keeping the diversity of individuals is helpful to avoid falling into local optimal solution, thus enhancing the global search ability. Compared with genetic algorithm and particle swarm optimization algorithm, SCSO has a simpler calculation process and robust optimization performance.

The first step of SCSO algorithm is population initialization; after initialization, the search for prey begins to find the optimal solution. In the SCSO algorithm, the parameter $\vec{r_{G}}$ decreases linearly from 2 to 0, as defined by Equation (11), where $S_{M}$ is assumed to be 2, $Iter_{c}$ is the current iteration number, and $Iter_{max}$ is the maximum number of iterations. Thus, in the initial iterations, sand cats move quickly; after half of the iterations, their movements become more intelligent. As with other metaheuristic algorithms, balancing exploration and exploitation phases is crucial. The SCSO uses the parameter $\vec{R}$ to achieve this balance. According to Equation (12), the transition between the two phases is balanced. Additionally, the formulation of Equation (13) aims to prevent the algorithm from falling into local optima. Parameter $\vec{r}$ determines the sensitivity range of each search agent. The core step of SCSO is updating the position of each search agent. Based on Equation (14), the position update of each search agent in every iteration depends on the best candidate position and its current position outside the sensitivity range. In Equation (14), $\vec{Pos_{bc}}$, $\vec{Pos_{c}}$, and $\vec{r}$ represent the best candidate position, current position, and sensitivity range, respectively. After the exploration stage (searching for prey), SCSO enters the exploitation stage (attacking prey). Equation (15) is used to calculate the distance between the best position and the current position of each search agent during the corresponding iteration. Assuming the sensitivity range is circular, the direction of each move is determined by a random angle θ selected by SCSO through a roulette wheel mechanism. The randomly chosen angle θ, ranging from 0 to 360 degrees, produces a cosine value between −1 and 1, thus achieving circular motion. In Equation (15), $\vec{Pos_{b}}$ and $\vec{Pos_{rnd}}$ represent the best position and random position, respectively [39].(11)$$\vec{r_{G}}=S_{M}-\left(\frac{S_{M}\times Iter_{c}}{Iter_{max}}\right)$$(12)$$\vec{R}=2\times \vec{r_{G}}\times rand\left(0,1\right)-\vec{r_{G}}$$(13)$$A\vec{r}=\vec{r_{G}}\times rand\left(0,1\right)$$(14)$${\vec{Pos}}_{\left(t+1\right)}=\vec{r}\times \left({\vec{Pos_{bc}}}_{\left(t\right)}-rand\left(0,1\right)\times {\vec{Pos_{c}}}_{\left(t\right)}\right)\;\vec{Pos_{rnd}}=\left|rand\left(0,1\right)\times {\vec{Pos_{b}}}_{\left(t\right)}-{\vec{Pos_{c}}}_{\left(t\right)}\right|$$(15)$${\vec{Pos}}_{\left(t+1\right)}={\vec{Pos_{b}}}_{\left(t\right)}-\vec{r}\times \vec{Pos_{rnd}}\times \cos\theta$$(16)$${\vec{X}}_{\left(t+1\right)}=\left\{\begin{array}{cc} {\vec{Pos_{b}}}_{\left(t\right)}-\vec{Pos_{rnd}}\times \cos\theta \times \vec{r} & \left|R\right|\leq 1\\ \vec{r}\times \left({\vec{Pos_{bc}}}_{\left(t\right)}-rand\left(0,1\right)\times {\vec{Pos_{c}}}_{\left(t\right)}\right) & \left|R\right|>1 \end{array}\right.$$

## 3. Experiment

### 3.1. F-P Cavity Interference Principle

The fiber Fabry–Perot cavity sensor is developed from optical Fabry–Perot (F-P) interferometer [40], which consists of two high reflectivity, strictly parallel optical plates spaced by a distance L. As shown in Figure 4, the reflectivities of the two optical plates is $\;\;R_{0}$ and $\;\;R_{1}$, respectively, while the refractive indices of the media inside and outside the F-P cavity are $\;n_{0}$ and $\;n_{1}$, respectively. The incident light intensity is $\;I_{0}$, the wavelength is λ, and the incident angle is θ.

For multi-beam interference generated by parallel flat plates, the optical path difference between adjacent reflected or transmitted light is calculated as follows:(17)ΔL=2nLcosθ

And the corresponding phase difference is(18)$$\delta =\frac{4\pi n_{0}L\cos\theta }{\lambda }$$

The principle of multi-beam interference can be applied to obtain the synthesized amplitude of reflected light, thereby obtaining the reflected light intensity:(19)$$I_{R}=\frac{R_{0}+R_{1}-2\sqrt{R_{0}R_{1}}\cos\delta }{1+R_{0}R_{1}-2\sqrt{R_{0}R_{1}}\cos\delta }I_{0}$$

Analysis of the Formulas (18) and (19) reveals that variations in the refractive index n, the incident angle θ, and the F-P cavity length L result in corresponding changes in the reflected light output $\;I_{R}$. Modulation can be achieved by altering these three parameters. Given that modulating the cavity length L of the F-P cavity is the simplest and most straightforward method, the EFPI sensor employed in this study is a F-P cavity based on the cavity length L modulation. As shown in Figure 5, the end face of optical fiber and aluminum film form an F-P microcavity. External impacts cause the aluminum film to vibrate, which will lead to a change in cavity length L. From Formulas (18) and (19), it can be seen that the reflected light intensity $I_{R}$ will also change. Therefore, by detecting the change in the intensity of the reflected light $I_{R}$, the detected impact signal can be demodulated.

### 3.2. Experimental Setup

The CFRP structure impact monitoring system is shown in Figure 6, comprising a broadband light source (Maxray Optoelectronics Technology Co., Ltd., Hefei, China), coupler (Shanze Jiye Technology Co., Ltd., Shenzhen, China), eight circulators (Qianhai Xunka Technology Co., Ltd., Shenzhen, China), four EFPI sensors, four fiber Bragg gratings (FBGs), four photodetectors (Hongyi Optoelectronics Technology Co., Ltd., Beijing, China), CFRP structural test pieces, a 1 J spring impact hammer (Qigong Instrument Equipment Co., Ltd., Shanghai, China), and a data acquisition system (Art Technology Co., Ltd., Shijiazhuang, China). The monitoring system is based on intensity demodulation. Taking Channel 1 as an example, the broadband light source emits light that is split into four channels by the coupler. One channel proceeds through optical circulator 1 to reach EFPI sensor 1. The reflected light from EFPI sensor 1 passes through optical circulator 5 and enters the FBG. The FBG 1 filters the interference signal reflected by EFPI sensor 1, and the narrowband reflected light $I_{R}$ obtained by FBG reflection will be received by the photodetector PD1. The working principle of intensity demodulation method is relatively simple, easy to implement, and has the advantages of low cost and fast dynamic response.

After establishing the impact monitoring system for the CFRP structure as depicted in Figure 6, an impact localization experiment was conducted on a 600 mm × 600 mm × 3 mm CFRP structure, with a 400 mm × 400 mm × 3 mm experimental area to minimize edge effects. As shown in Figure 7, the lower left corner of the monitored area was designated as the coordinate origin Q1 (0 mm, 0 mm), with the horizontal axis as the *X*-axis and the vertical axis as the *Y*-axis, forming a two-dimensional rectangular coordinate system.

Four EFPI sensors were adhered to the four corners of the monitored area using epoxy resin glue. The entire impact monitoring area was divided into 17 × 17 grids, and each small grid is 2.5 cm × 2.5 cm in size. The intersections of these grids represented impact points, totaling 285 (excluding sensor positions). The impact test was performed using a spring hammer with an impact energy of 1 J, with the sampling rate set to 10 M/s, and three impacts were performed at each position, resulting in a total of 855 datasets.

According to Figure 6, an EFPI signal acquisition system based on intensity demodulation is built, as shown in Figure 8. Among them, 1 is amplified spontaneous emission (ASE) broadband light source; 2 is 1 × 4 coupler; 3~10 are circulators; 11 is a multi-channel voltage stabilized source, which provides a constant voltage of 12 V for photoelectric detectors; 12–15 are EFPI sensors; 16–19 are FBGs; 20~23 are photoelectric detectors; 24 is a data acquisition card; 25 is a computer.

## 4. Feature Extraction

### 4.1. Data Preprocessing

When the impact hammer strikes the CFRP structure, it causes vibrations that generate impact stress waves. These waves propagate through the surface of the CFRP structure, and the closer the EFPI sensor is to the impact point, the shorter their transmission time. Consequently, the time of the impact signal received by the EFPI sensor is closely related to the relative location between the impact point and the monitoring sensor. Each EFPI sensor is employed to capture the time domain response signal characteristics at various locations following the impact.

Taking the response signals collected at impact points B3 (50 mm, 375 mm) as an example, the Daubechies wavelet function (abbreviated as DbN) is utilized due to its advantages of good regularity, smooth vanishing moment, strong localization ability in the frequency domain, and effective frequency band decomposition. Specifically, the Db8 wavelet function is employed to decompose the impact signal into eight layers [41]. Subsequently, the low-frequency scale a8 is used to reconstruct the narrow-band components generated by Lamb waves. Energy–entropy ratio endpoint detection is then performed on the reconstructed impact signal to determine the arrival time of the response signal at the monitoring sensor. The impact signals from four EFPI sensors are shown in Figure 9.

The energy–entropy ratio sequence of the impact signals detected by four EFPI sensors and the corresponding extraction process of the arrival time of the impact point are shown in Figure 10. The energy–entropy ratio method can highlight the difference between the impulse signal and noise, making the identification of the endpoint position of the impact signal less sensitive to the selection of threshold. In this study, the length of the short time window was set to 200, and the threshold was set to 0.1 mV. The endpoint positions of the response signals detected by the four EFPI sensors using the energy–entropy ratio method were 0.04941 s, 0.04973 s, 0.04995 s, and 0.04971 s, respectively, and are marked with red lines. It can be easily observed that the impact signal first reaches EFPI 1, almost simultaneously reaches EFPI 2 and EFPI 4, and finally reaches EFPI 3. This is consistent with the distance from the impact point to the sensors, confirming that the arrival times extracted by the energy–entropy ratio method can serve as features for impact localization.

### 4.2. Sensitivity Analysis

The contour map of the normalized arrival time differences for impact signals between EFPI 1 and the other EFPI sensors is presented in Figure 11. In order to mitigate discrepancies in sensor arrival times caused by location or other factors, we standardize these time differences for a unified comparison across sensors. This normalization process reflects the relative delay in signal propagation and considers the standardized time difference as the characteristic quantity of the impact signals. The specific formula is as follows:(20)$$\Delta T_{1}=T_{1}-T_{2},\;\Delta T_{2}=T_{1}-T_{3},\;\Delta T_{3}=T_{1}-T_{4}$$

The normalized arrival time difference is expressed as follows:(21)$$\Delta T_{norm}=\frac{\Delta T-\min(\Delta T)}{\max(\Delta T)-\min(\Delta T)}$$

Figure 11 illustrates that the time difference between impact signals varies significantly with the distance from the impact point to the monitoring EFPI sensors. When the impact point is close to one sensor and distant from another, the time difference is considerable. Conversely, if the impact point is equidistant from two sensors, the time difference is negligible. This demonstrates that EFPI sensors at different locations exhibit high sensitivity to the location of impact point. Consequently, the time difference extracted using the energy–entropy ratio method be used as the feature for machine learning models to locate the impact on CFRP structures.

## 5. Impact Localization

### 5.1. Establishment of Localization Model

In the experiment, 855 sets of data were obtained and divided into the training set, verification set, and test set. During the model training process, the difference between the arrival time of EFPI1 and the other three EFPI sensors $\Delta T_{1},\Delta T_{2},\Delta T_{3}$ is taken as the input, and the actual impact coordinates are taken as the output.

The CNN-BIGRU-Attention model is established with 800 sets of data for training, 40 sets of data for verification, and 15 sets of data for testing. The accuracy of the model is evaluated by calculating the average distance error between the actual and predicted impact point locations. The main steps to establish the prediction model are as follows:

Step 1: Partition and standardization of data

A random selection of 800 samples is used to form the training set. From the remaining 55 samples, another 40 are randomly chosen for the validation set, leaving the final 15 samples to serve as the test set. Then, the input features of training set and test set are normalized to the range of [0, 1]. This step eliminates the dimensional differences between different features, enhancing model stability and accelerating convergence during training.

Step 2: Establish the deep learning model

The flowchart of the CNN-BIGRU-Attention model employed in this paper is shown in Figure 12.

The CNN module comprises two feature extraction layers, a Flatten layer, and a Fully Connected (FC) layer. Each feature extraction layer consists of four modules: Batch Normalization (Batch Norm), Rectified Linear Unit (ReLU), and Max Pooling. The convolution kernels of the two CONV layers have a size of 3, with a step size of 1. The Batch Norm layer standardizes the input of each layer, accelerating training and enhancing stability. The ReLU layer introduces nonlinearity, aiding the model in learning complex relationships and mitigating gradient disappearance. Max Pooling reduces the dimensionality of features by downsampling the output of the convolution layer, thereby improving training efficiency.

The Flatten layer converts the output of the convolution layer into one-dimensional vectors, preparing the data for input into the subsequent FC layer. The FC layer integrates the previously extracted features to improve the nonlinear expression ability of features. The BIGRU module, paralleled with the CNN, contains two opposing GRU modules with identical parameters and 35 hidden units.

The concatenation layer integrates features from the CNN and BIGRU, facilitating feature fusion and information integration. This enhances model performance and expressiveness. Additionally, the attention mechanism allows the model to learn more intricate function mappings by dynamically focusing on various parts of the input, thereby improving its expressive capacity. Simultaneously, it effectively captures relationships between different positions in the input sequence, enabling the model to prioritize information that is distant from the current unit, thus better handling long sequence data.

The CNN-BIGRU-Attention model works as follows:The normalized dataset undergoes format conversion to form a three-dimensional matrix;The convolution layer extracts local features of the input sequence data through multi-layer convolution operations;The pooling layer reduces dimensionality and compresses the data to lower computational complexity;BIGRU captures long-term dependencies in time series by processing bidirectional sequence information;The concatenation layer combines the outputs of the convolutional neural network and the Bidirectional Gated Recurrent Unit (BIGRU);The attention mechanism module assigns different weights to each position by calculating the similarity between positions;Finally, after processing by the fully connected (FC) layer and an activation function, the output is passed to the regression layer. This regression layer maps the final output of the neural network to the range required by the regression task, generating the predicted localization coordinates.

Step 3: Optimize the model by using sand cat swarm optimization algorithm

The specific steps for optimizing the model using the SCSO algorithm are as follows:Set initial parameters: initial population size is set to 20, and maximum evolutionary generation is set to 20. The parameters to be optimized include learning rate, number of neurons in BIGRU, attention mechanism key value, and convolution kernel size. The initial learning rate is set to 0.005, and the upper and lower bounds are 0.01 and 0.001, respectively. The initial number of neurons in BIGRU is set to 35, and the upper and lower bounds are 50 and 10, respectively. The initial attention mechanism key value is set to 30, and the upper and lower bounds are 50 and 2, respectively. Initial convolution kernel size is set to 3, and the upper and lower bounds are 10 and 2, respectively.Calculate the fitness of each individual in the population;Record the current best individual, select the individual with higher fitness for evolution, and generate the next generation;Renew the population by replacing poorly performing individuals with newly generated ones, ensuring gradual convergence to the optimal solution;Iterate steps 2–4 until the termination condition is met.Retrieve the best parameters and output the optimal solution.

Step 4: Prediction of impact point

The impact samples from the test set were fed into the CNN-BIGRU-Attention model, which was optimized using the sand cat swarm optimization algorithm and the impact samples in the validation set. Subsequently, the model generates predicted coordinates for the impact points in the test set. Ultimately, the average coordinate error was calculated using the obtained results.

The formula for calculating the average distance error between predicted and actual locations is expressed as follows:(22)$$\bar{d}=\sum_{1}^{n}\sqrt{{\left(x_{i}+x_{i}^{*}\right)}^{2}+{\left(y_{i}+y_{i}^{*}\right)}^{2}}$$
where $x_{i}$ represents the actual abscissa value, $x_{i}^{*}$ represents the predicted abscissa value, $y_{i}$ represents the actual ordinate value, $y_{i}^{*}$ represents the predicted ordinate value, and n represents the number of predicted coordinates.

### 5.2. Analysis of Localization Results

In order to validate the reliability and precision of the impact localization method based on CNN-BIGRU-Attention model, Figure 13 and Table 1 illustrate the impact localization results alongside prediction errors across various models. According to Table 1, the maximum localization errors of CNN, CNN-GRU, CNN-BIGRU, and CNN-BIGRU-Attention models are 19.92 mm, 17.25 mm, 16.78 mm, and 11.82 mm, respectively. Correspondingly, their minimum localization errors based on the four models are 11.67 mm, 8.89 mm, 8.25 mm, and 5.01 mm. The average errors are 15.73 mm, 12.66 mm, 11.30 mm, and 8.03 mm, respectively. The results collectively demonstrate that the CNN-BIGRU-Attention model-based impact localization method outperforms in terms of localization accuracy. Additionally, as the structural complexity of the model rises, the average running time for impact localization using CNN, CNN-GRU, CNN-BIGRU, and CNN-BIGLU Attention models is 2.659 s, 3.253 s, 3.701 s, and 4.426 s, respectively. Compared with the localization results of impact on CFRP plate structures using FBG sensors, the localization accuracy of this system based on EFPI sensors is also higher [42].

## 6. Conclusions

In this paper, an impact localization system for CFRP structures by using EFPI sensors demodulated at high speed was established, and an impact localization method based on CNN-BIGRU-Attention model for the CFRP structure is proposed. Firstly, the low-frequency narrowband signal component of the impact signal generated by Lamb waves is extracted using wavelet packet decomposition to mitigate dispersion effects. Then, an endpoint detection method based on the energy–entropy ratio algorithm is used to extract the time at which the impact signal reaches each EFPI sensor and verify its sensitivity. Secondly, the arrival time differences in impact signals detected by different EFPI sensors were collected and used as features to train and optimize the CNN-BIGRU-Attention model for impact localization. Finally, the impact localization methods based on different models were validated on a 400 mm × 400 mm × 3 mm CFRP plate. By comparing the prediction results of different models, the proposed impact localization method for the CFRP structure based on CNN-BIGRU-Attention was proved to achieve superior localization accuracy.

## Figures and Tables

**Figure 1 sensors-25-01091-f001:**
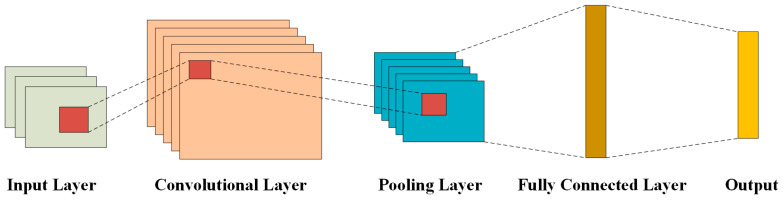
CNN structure schematic.

**Figure 2 sensors-25-01091-f002:**
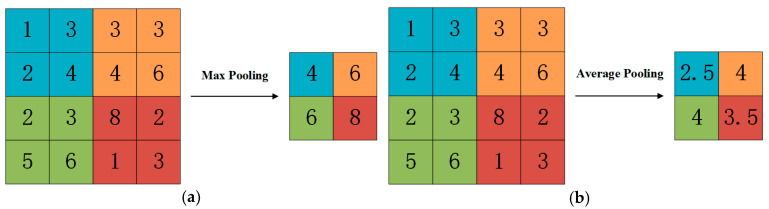
Schematic diagram of pooling operations: (**a**) maximum pooling; (**b**) average pooling.

**Figure 3 sensors-25-01091-f003:**
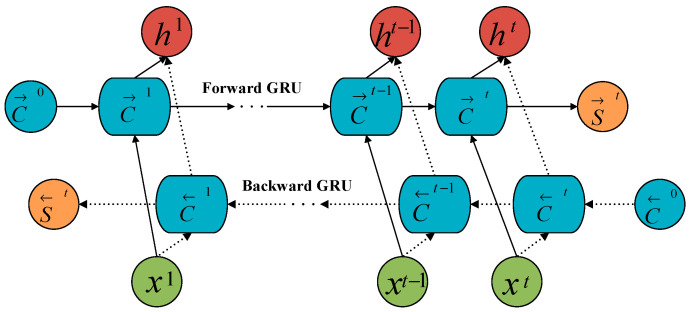
BIGRU network structure model.

**Figure 4 sensors-25-01091-f004:**
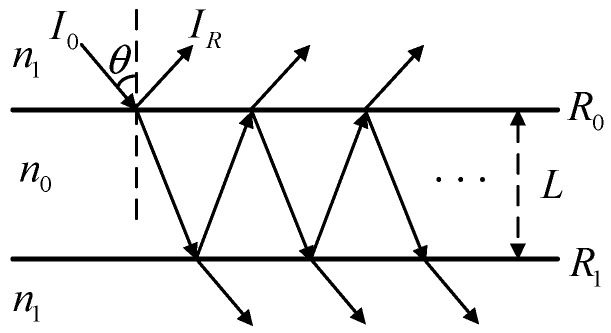
Schematic diagram of F-P cavity interference.

**Figure 5 sensors-25-01091-f005:**
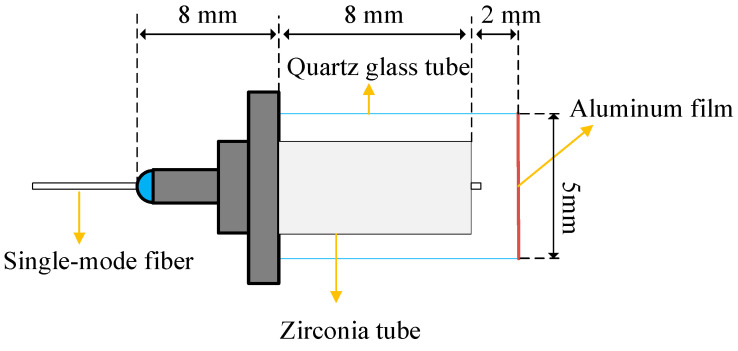
Structure diagram of EFPI sensor.

**Figure 6 sensors-25-01091-f006:**
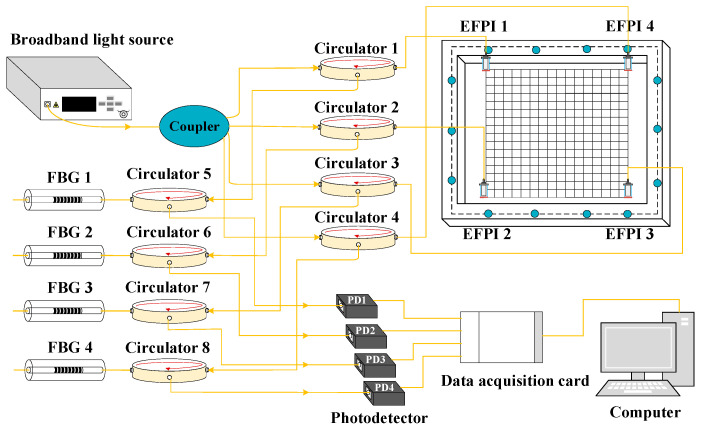
EFPI system demodulated based on intensity method.

**Figure 7 sensors-25-01091-f007:**
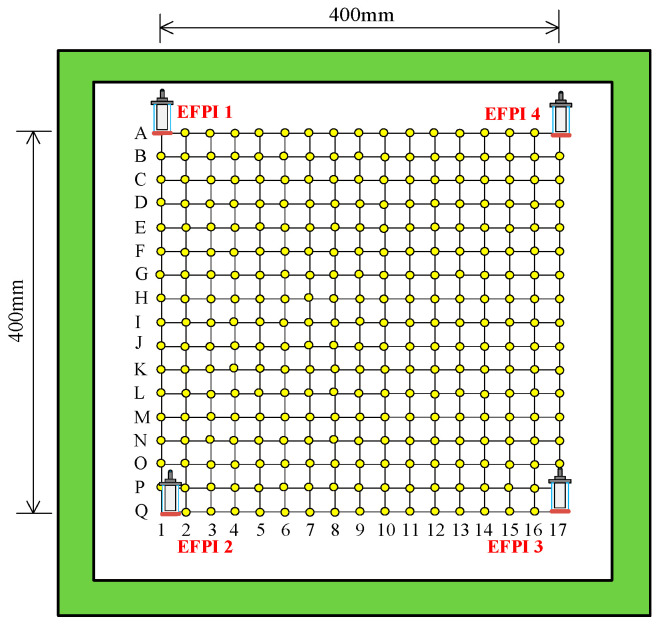
Schematic diagram of impact monitoring area division.

**Figure 8 sensors-25-01091-f008:**
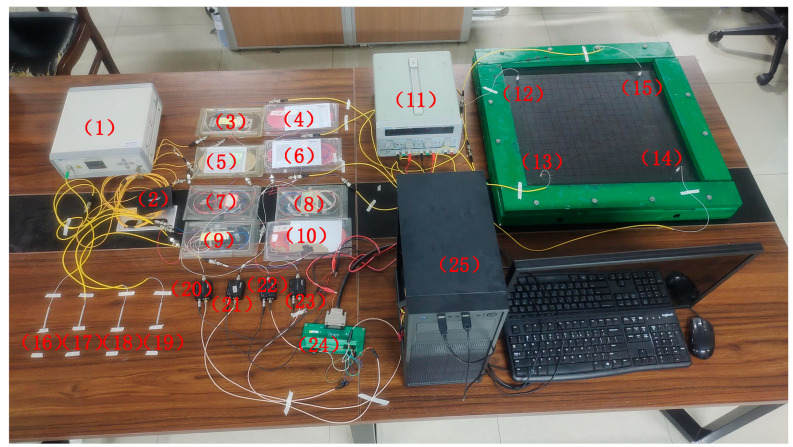
Impact localization system of CFRP structure based on EFPI sensors.

**Figure 9 sensors-25-01091-f009:**
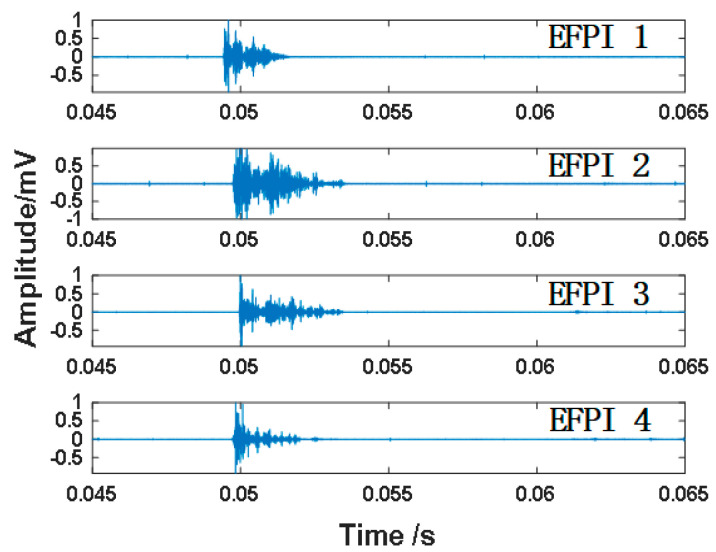
Impact signals detected by different EFPI sensors.

**Figure 10 sensors-25-01091-f010:**
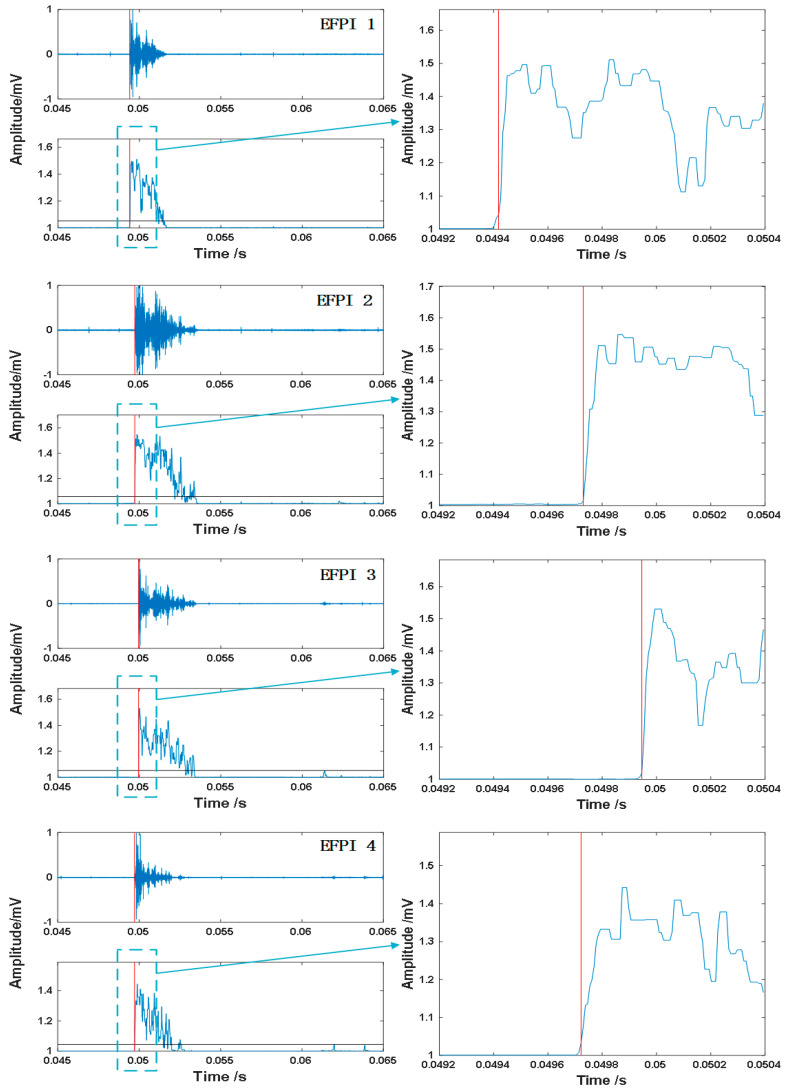
Impact signal and corresponding energy–entropy ratio.

**Figure 11 sensors-25-01091-f011:**

Normalized time difference in impact signal arrival between different EFPI sensors. (**a**) Normalized time difference in impact signal arrival between EFPI 2 and EFPI 1. (**b**) Normalized time difference in impact signal arrival between EFPI 3 and EFPI 1. (**c**) Normalized time difference in impact signal arrival between EFPI 4 and EFPI 1.

**Figure 12 sensors-25-01091-f012:**
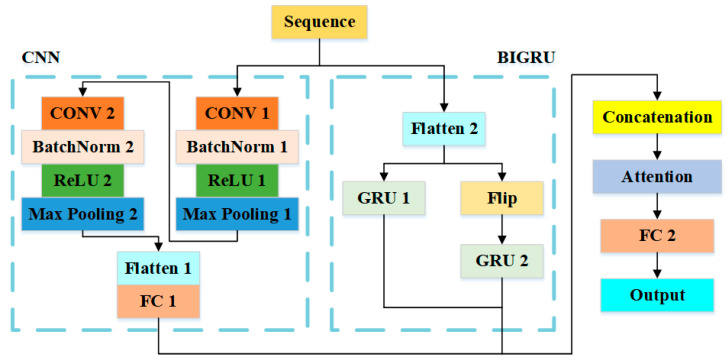
Flow chart of impact localization method based on CNN-BIGRU-Attention.

**Figure 13 sensors-25-01091-f013:**
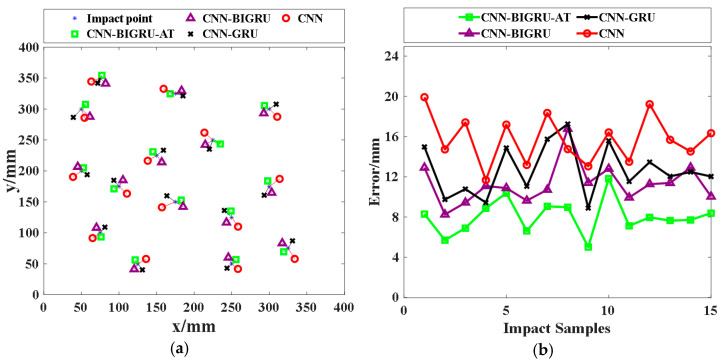
(**a**) Impact localization results; (**b**) impact localization errors.

**Table 1 sensors-25-01091-t001:** Comparison of impact localization result.

Impact	Impact Coordinate (mm)	CNN Algorithm		CNN-GRU Algorithm		CNN-BIGRU Algorithm		CNN-BIGRU-Attention Algorithm	
		Predict Coordinates (mm)	Error (mm)	Predict Coordinates (mm)	Error (mm)	Predict Coordinates (mm)	Error (mm)	Predict Coordinates (mm)	Error (mm)
1	(175, 150)	(157.106, 141.247)	19.920	(163.726, 159.847)	14.968	(185.262, 142.160)	12.914	(182.861, 152.650)	8.296
2	(50, 200)	(38.865, 190.362)	14.727	(57.655, 193.975)	9.741	(45.123, 206.653)	8.249	(52.680, 205.027)	5.696
3	(175, 325)	(159.364, 332.655)	17.410	(185.153, 321.375)	10.781	(183.257, 329.561)	9.433	(168.109, 324.683)	6.899
4	(250,50)	(258.265, 41.756)	11.674	(243.764, 42.925)	9.431	(245.514, 60.133)	11.081	(255.553, 56.919)	8.871
5	(250,125)	(258.462, 110.036)	17.191	(240.236, 136.217)	14.871	(242.995, 116.674)	10.881	(248.995, 135.334)	10.383
6	(75,100)	(64.888, 91.546)	13.180	(81.236, 109.133)	11.059	(70.053, 108.264)	9.631	(76.103, 93.481)	6.611
7	(300,175)	(313.763, 187.137)	18.350	(293.237, 160.783)	15.744	(303.265, 164.797)	10.713	(297.923, 183.815)	9.057
8	(50,300)	(54.160, 285.846)	14.752	(39.237, 286.522)	17.249	(61.236, 287.543)	16.776	(55.365, 307.191)	8.972
9	(75,350)	(63.196, 344.456)	13.041	(71.736, 341.729)	8.891	(82.154, 341.133)	11.393	(77.321, 354.439)	5.009
10	(225, 250)	(213.654, 261.861)	16.414	(220.237, 235.165)	15.581	(214.824, 242.237)	12.800	(234.860, 243.484)	11.819
11	(125,50)	(135.985, 57.841)	13.497	(131.137, 40.137)	11.532	(120.232, 41.279)	9.940	(121.593, 56.255)	7.122
12	(325,75)	(333.824, 57.924)	19.221	(330.863, 87.116)	13.460	(317.265, 83.198)	11.271	(319.220, 69.517)	7.966
13	(100, 175)	(110.541, 163.388)	15.683	(93.133, 184.896)	12.045	(105.270, 185.091)	11.385	(93.431, 171.093)	7.643
14	(150, 225)	(138.256, 216.463)	14.519	(159.165, 233.467)	12.478	(156.880, 214.053)	12.930	(145.224, 231.053)	7.710
15	(300, 300)	(310.563, 287.538)	16.336	(309.169, 307.790)	12.031	(292.685, 293.132)	10.034	(293.622, 305.496)	8.389
Max. error			19.920		17.249		16.776		11.819
Min. error			11.674		8.891		8.249		5.001
Avg. error			15.728		12.656		11.295		8.030

## Data Availability

The data are available from the corresponding author on reasonable request.

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
