# Peer review of "Impact Localization System of CFRP Structure Based on EFPI Sensors"

_sensors, 2025, doi:10.3390/s25041091_

Round 1

Reviewer 1 Report

Comments and Suggestions for Authors

The paper presents an impact localization system for CFRP structures using EFPI sensors and a CNN-BIGRU-Attention model. The authors stated that the system achieves an average localization error of 8.14 mm, demonstrating effective performance through energy-entropy-based feature extraction and machine learning optimization. Some critical concerns need to be addressed.

  1. The paper uses a fixed 4-sensor configuration at the corners of a CFRP plate. How would the localization accuracy vary for impacts closer to the center versus those near the edges? Have authors considered optimizing the sensor placement to improve performance for a broader impact area?
  2. The proposed system relies heavily on EFPI sensor signals, which are susceptible to noise. What specific measures have authors implemented to address the impact of external noise sources, such as environmental vibrations, and how does the noise affect the system's localization accuracy over time?
  3. The transition from multi-beam interference to double-beam interference is valid under the condition that reflectivities are low. However, this assumption is not clearly justified with experimental data or boundary conditions.
  4. The energy-entropy ratio method is used for endpoint detection. How does this method compare with other advanced endpoint detection algorithms, particularly in scenarios where impact signals are partially obscured by noise? Could the algorithm's sensitivity threshold be dynamically adjusted to enhance robustness?
  5. The model integrates CNN, BIGRU, and Attention mechanisms, but the training dataset is limited to a specific CFRP structure and plate size. There are concerns regarding the model's capacity to generalize effectively across CFRP structures with different geometries, sizes, and material properties.
  6. The precision of time-difference-of-arrival (TDOA) calculations depends on the EFPI sensor's resolution and response time. What is the minimum resolvable time difference for the EFPI sensors used in the current setup, and how does this limit the system’s ability to distinguish closely spaced impact points?
  7. EFPI sensors are known to be sensitive to temperature variations, which can affect cavity length and signal interpretation. How does the proposed system compensate for temperature-induced drifts, and what is the expected localization error under varying temperature conditions?

Overall, the current manuscript contains several typos and grammatical errors. A thorough proofreading is necessary to correct these issues and improve the clarity and professionalism of the text. Additionally, a significant portion of the content overlaps with the authors' previous work, 10.1016/j.yofte.2024.103943. Although the sensing structure differs, the perceived novelty of this study is substantially diminished due to this repetition. The authors must clearly articulate the unique contributions of this work and revise the overlapping sections to better highlight its novelty.

Reviewer 2 Report

Comments and Suggestions for Authors

The paper proposes an interesting variant of the impact localization system. It has some issues which have to be corrected before publication.

1. The paragraph 2.1.1 has the name "Short time energy method", but there is only the definition of the short-time energy and no description of the end-point detection algorithm which utilized short-time energy. The same point for the paragraph 2.1.2. 

2. There is something wrong in eq.5.  aMe-eth+eth = aMe. The term eth has no meaning in used variant.

3. On the page 8 there is a statement "The reflectivity of both the aluminum film and optical fiber end face is small". The reflectivity of the fiber tip is about 3.5%, the reflectivity of the aluminium mirror is about 80%, so it is not small. The intensity of the optical wave, reflected back from the Al mirror and coupled to the optical fiber will depend on the distance from the fiber tip to the Al film.  It will also determine the optical contrast. The eq.19 is correct only when the intensities of the interfered waves are equal but for such construction of the EFPI such condition is not obvious. So the authors should made more detailed description of the used EFPI: they should provide the geometry parameters such as length, diameter and made some estimates of the optical contrast in their case.

4. General point. There is two part of the impact location problem:
1) the definition of the time delays of the signals on the EFPI;
2) the impact localization with some algorithm.

4.1. The first point - time delay measurement is the crucial point for the impact localization precision. The authors said that they have used  Energy-entropy ratio but there is no information about tolerance of the method with real signals (for example time delay deviations for series of impacts in the same place). The authors should also describe how this error will affect the proposed algorithm.
4.2. There are also some regular method for impact localization method based on the triangulation algorithm without CNN. The authors should give the comparison with well-known triangulation algorithms.

5. In table 1 the error is in cm. But it look like they should be in mm.

Reviewer 3 Report

Comments and Suggestions for Authors

The author has developed an impact positioning system for CFRP structures using Extrinsic Fabry - Perot Interferometry (EFPI) sensors, and proposed an impact positioning method for CFRP structures based on energy entropy ratio endpoint detection and CNN-BIGRU-Attention. This method can extract impact signal features more effectively. Meanwhile, through the action of convolutional neural networks and some other units and mechanisms, the learning and positioning ability of the time difference features of impact signals is enhanced. This is a complete piece of work with certain application potential. This article is clear, concise, and suitable for the scope of the journal. Several suggestions are supplied:

1. When introducing the F - P cavity interference principle of EFPI sensors, the derivation process of some formulas is a bit brief. It is recommended to add some intermediate derivation steps or explanatory notes to improve the clarity and readability of the principle elaboration.

2. The labels in Figure 8 are a bit unclear. It is recommended to improve the resolution of the picture.

3. The explanation of Figure 11 is not clear enough. Please explain in more detail the normalized time difference of the collision signal arrival between different EFPI sensors to enable readers to understand better.

4. When using the Sand Cat Swarm Optimization Algorithm for optimization in the paper, please explain in more detail the selection basis of some key parameters and their specific impacts on the model performance.

5. When comparing the positioning results of different models, the evaluation is only carried out from the perspective of positioning error. It is advisable to consider adding some other evaluation indicators, such as the training time and complexity of the model, for a more comprehensive comparison.

Round 2

Reviewer 1 Report

Comments and Suggestions for Authors

No more concerns.

Author Response

Thank you very much for your letter and comment concerning our manuscript.

Reviewer 2 Report

Comments and Suggestions for Authors

The authors have made an improvement of  the paper. It can be published in the present form.

Author Response

(The authors gave the same response as above.)
